# The Influence of Recommendation of Medical and Non-Medical Authorities on the Decision to Vaccinate against Influenza from a Social Vaccinology Perspective: Cross-Sectional, Representative Study of Polish Society

**DOI:** 10.3390/vaccines11050994

**Published:** 2023-05-17

**Authors:** Tomasz Sobierajski, Piotr Rzymski, Monika Wanke-Rytt

**Affiliations:** 1The Center of Sociomedical Research, Faculty of Applied Social Sciences and Resocialization, University of Warsaw, 26/28 Krakowskie Przedmieście Str., 00-927 Warsaw, Poland; 2Department of Environmental Medicine, Poznan University of Medical Sciences, 60-806 Poznań, Poland; rzymskipiotr@ump.edu.pl; 3Integrated Science Association (ISA), Universal Scientific Education and Research Network (USERN), 60-806 Poznań, Poland; 4Department of Pediatrics with Clinical Assessment Unit, Medical University of Warsaw, 63a Żwirki i Wigury Str., 02-091 Warsaw, Poland; monika.wanke@uckwum.pl

**Keywords:** nurse, physician, pharmacist, sociology, vaccine opponent, vaccine supporter

## Abstract

Vaccination against seasonal flu is crucial to prevention of illness in modern societies. The level of influenza vaccination in Poland is low and, for many years, has hovered around a few percent of the general population. For this reason, it is crucial to understand the reasons for such a low level of vaccination and to assess the influence of medical and social authorities on the decision to vaccinate against influenza from the perspective of social vaccinology. For this purpose, a representative survey was conducted in 2022 among adult Poles (*N* = 805), orchestrated with the CAWI technique based on the author’s questionnaire. The most significant authority in the context of influenza vaccination is held by physicians, especially among the oldest part of the population, over 65 years of age—in this group, 50.4% of respondents declare a very high level of respect for physicians on the issue of recommended influenza vaccination (*p* < 0.001), and the second-highest authority group for which seniors have respect in the aspect of influenza vaccination is pharmacists (*p* = 0.011). It was also shown that pharmacists have more authority on the issue of influenza vaccination than nurses, especially in the group that declared themselves opponents of vaccination (*p* < 0.001). The survey indicates the need to strengthen the authority of physicians and pharmacists regarding influenza vaccination, and, in the case of pharmacists, the need for changing the law to allow them to qualify for influenza vaccination.

## 1. Introduction

Vaccination can decrease influenza burden by averting infections, medical visits, hospitalizations, and deaths [1,2,3]. Public attitudes toward influenza vaccination in the European Union (E.U.) vary significantly from country to country [4]. According to a recent survey, over 80% of Portuguese, Spaniards, Swedes, and Danes are convinced that such vaccination is beneficial, while this belief is much lower than 60% among Bulgarians, Slovaks, and Latvians. In Poland, 78% of investigated individuals declared that influenza vaccination is safe, and approximately 70% declared that it is also effective and important [4]. Although this is a marked increase compared to the 2018 study, in which less than 60% of surveyed Poles indicated were convinced that flu vaccination is safe and needed, the vaccination rate in Poland remains very low. Usually, its coverage does not exceed 5% of the general population, and rates as one of the lowest in Europe [5,6,7]. The highest vaccine uptake rates are noted in individuals over 65 years. However, in the 2019/2020 epidemic season, the uptake in this group was only 10.4%, much lower than the average coverage in the E.U. member states (44%) [5]. At the same time, the influenza vaccine coverage among Polish healthcare workers also remains low [8]. This is despite the fact that during the COVID-19 pandemic, several studies have suggested that influenza vaccination may also reveal cross-protective effects, translating into lower rates of SARS-CoV-2 infection, subsequent hospitalization, and deaths, as well as improved adaptive responses of the immune system to the SARS-CoV-2 challenge [9,10,11]. One could therefore expect that these observations could improve the vaccination uptake in Poland, particularly during autumn-winter 2020, due to the absence of COVID-19 vaccine availability. This justifies the need to understand various factors that may play a role in such a low interest in influenza vaccination in Poland. The representative study by Samel-Kowalik et al., conducted among Poles in 2020, found that among those aged over 60, low levels of religious practices and internet use were significantly associated with positive attitudes toward influenza vaccination [12].

As vividly demonstrated during the COVID-19 pandemic, immunization is a social phenomenon [13,14]. To explain low or high immunization rates, the nature of anti-vaccine attitudes, and the increasing number of refusals to vaccinate young children, it is necessary to treat immunization as a social activity. While the vaccine itself is a medical product, the phenomenon of immunization and the decision to vaccinate or not to vaccinate against a disease (with a series of complex fears and motivations) are social phenomena and social actions that should be explained using the theoretical grid of social science. This is the task of social vaccinology, which aims to study, analyze, and describe social influences on people’s vaccinological attitudes [15]. Among other things, social vaccinology, as an interdisciplinary scientific research path, focuses on analyzing the phenomenon of vaccination at the interface of the biomedical and holistic models of health. 

On the one hand, vaccination is a medical action that is part of the biomedical model. On the other hand, it is a process for maintaining health status, subject to social reflection, referring to the holistic model of health. Since the end of the 19th century, there has been a biomedical model of health in medicine in which the focus is a disease and the fight against it. This model was based on germ theory, or the theory of infectious diseases, out of which modern vaccinology grew [16]. The biomedical model is not infrequently stereotyped, and attempts have been made to replace it with the holistic model, which grew out of the World Health Organization’s definition of health (1946), which states that health does not merely mean the absence of disease but is physical, mental, and social well-being [17]. This definition, although vague concerning concepts of social or mental well-being and thus sometimes difficult to measure, points out that a person’s health is influenced by many more elements than just the physical state of his or her body. Thus, the holistic concept of health corresponds perfectly with the modern approach to health as an interconnected system of vessels in which both the course of illness and treatment are influenced by the social environment and mental state [17]. 

Regardless of the paradigmatic considerations of the dominance of one model over the other in modern science, it should be emphasized that both models of health are applicable in medicine and vaccinology. The biomedical model draws attention to the value of medical achievements. In contrast, the holistic model, sometimes called the social model, points out that specific treatment models can have different effects depending on the patient’s mental state and social environment. It is related to the salutogenic thinking of Antonovsky [18], who drew attention to the “sense of coherence” associated with health, i.e., considering, in addition to planning health policy, the many aspects that affect the “built-in” immunity of a given social group, or, closer to Antonovsky’s thinking, class [18]. For this reason, it is essential to pay attention to demographic and social factors that can significantly impact attitudes toward vaccination when analyzing social attitudes toward vaccination [19,20,21,22]. 

One element of social influence on people’s attitudes, including attitudes toward vaccination, is authority. In psychosocial terms, authority bestows respect and trust on a specific person or group [23]. The basis for generating trust in people is the compilation of qualities that an authority figure should possess. Some authorities are universally and culturally recognized (e.g., physicians), while others are individually recognized (e.g., friends) [24,25]. Someone may become an authority because of high professional qualifications, someone else because of professed moral values.

Nonetheless, authority can strongly influence decision-making, whether we view it in terms of internal inclinations or institutionalized authority [26]. Regarding vaccinations, it is socially recognized that those who work in health care (physicians, nurses) are the epistemic authorities because they are familiar with medicine [27,28,29,30]. Some studies emphasize the significant role of pharmacists in preventive health care, since access to them and their knowledge is often much more accessible than to physicians and nurses [31,32]. Moreover, trust in them regarding vaccination can influence patients’ decisions to vaccinate against influenza. Individual authorities, not recognized by the public, may also influence one’s actions, including those related to preventive health care, such as vaccination. These include friends or family members who can be classified as deontic authorities by their function [33,34]. 

Given the above considerations related to the nature of vaccination as a social action and the influence of authority figures on the decisions people make, the main objective of our study was to examine the influence of medical, socially recognized authorities such as physicians, nurses, pharmacists, and subjective, individual, personal authorities such as family and friends on people’s attitudes toward getting vaccinated against influenza. During the study, we sought answers to the following research questions: To what extent would a physician’s or nurse’s opinion influence your decision to vaccinate against influenza?To what extent a friend’s/family member’s opinion would influence a respondent’s decision to be vaccinated against influenza?To what extent would a pharmacist’s opinion influence flu vaccination?How does the respondent’s attitude toward vaccination correlate with trust in medical authority regarding influenza vaccination?Do sex and age affect the trust in medical authority regarding influenza vaccination?

## 2. Materials and Methods

### 2.1. Implementation of the Study and Sample Size

The survey was conducted in Poland on October 2022, at the beginning of the influenza epidemic season, using computer-assisted web interviewing (CAWI). The representative sample of 805 Poles was stratified into demographic layers: sex, age, education, place of residence, and income. Object representativeness assumes that, apart from the layers mentioned above, no other significant variables could influence the results obtained in the study. 

### 2.2. Operationalization of the Concept of Authority

People’s decisions to vaccinate or not to vaccinate against influenza can be influenced by various factors [35,36,37]. These can be divided into personal sources of information, i.e., directly acquiring knowledge during an encounter with another person, and impersonal sources of information, i.e., all the information we acquire from traditional and social media. The present study explored the potential influence of personal authorities on decisions toward influenza vaccination. Based on the results of previous analysis, which shows that when it comes to personal sources of information about vaccination, Poles place the most trust in the following groups, in order: physicians, nurses, friends, and family [38], we decided to include these groups in our study. In addition, we included pharmacists in the group of medical authorities. This is because in Poland, according to changes in Polish law in 2021, the qualification for influenza vaccination can be issued by a physician or pharmacist, although only the former can issue the prescription, which is pivotal to purchasing the vaccine [39]. Similarly, nurses can qualify for vaccinations and perform them, but only based on a prescription issued by a physician. 

We realize that proposed health decisions can be influenced by, for example, celebrities and influencers, who are authorities for many people. However, this group has been excluded from the present study because its knowledge is mediated by the media and not based on personal contact.

### 2.3. Questionnaire Design

The questionnaire used in this study was initially designed for this study, following the latest sociological and methodological knowledge of the authors. The questionnaire consisted of 13 questions: 5 metric questions and 8 factual questions. The former asked about sex, age, education, place of residence, and income. The factual questions considered attitudes toward vaccinations and attitudes toward particular authorities. Each question was closed. For the metric questions, the criteria were adjusted according to the scope of the question, such as the size of the locality or educational level. For the factual questions, respondents were asked to what extent a person’s opinion (implicitly an authority figure) would influence influenza vaccination. To this end, a Likert-like scale 5-degree scale was used, where one end of the scale meant “very low influence”, and the other end indicated “very high influence”. In addition, in the question about attitudes toward vaccination: “What is your attitude towards vaccinations in general?”, a 4-degree scale was used. Respondents were asked to assign themselves, according to their assessment, to one of 4 categories: strong vaccine opponent, moderate vaccine opponent, moderate vaccine supporter, and strong vaccine supporter. 

The questionnaire underwent a rigorous validation process, beginning with an ad hoc examination and subsequent revision by qualified interdisciplinary experts. It was then tailored to the specific research technique to be employed. A thorough evaluation was conducted to ensure the effectiveness of the questionnaire, including a pilot study involving 30 participants selected at random. The pilot group was not included in the final study. After carefully reviewing feedback and making necessary adjustments to the questionnaire, it was technically adapted and integrated into the research company’s panel. Finally, it was disseminated to randomly selected participants via a link. The translated version of the questionnaire (Polish → English) is available as online Appendix A.

### 2.4. Statistical Analysis

The statistical analysis was carried out using IBM SPSS Statistics v. 28.0.1.0 (IBM Corp., Armonk, NY, USA). The data for all outcomes were recorded for all participants. To evaluate the relationship between variables, the Chi-square test was used. The Kruskal–Wallis test was employed to analyze questions that utilized the Likert scale. Responses to questions were presented with the total number of respondents (*n*) and frequency percentages of sub-groups (%). A statistically significant *p*-value was considered to be less than 0.05 for all analyses. 

### 2.5. Ethical Considerations

The research company is a member of the European Society for Opinion and Marketing Research (ESOMAR) and offers assurance for the ethical execution of the study and the safeguarding of participants’ data. All survey participants gave informed consent to participate in the study. The study was conducted following all ethical rules of Poland and the European Union relating to the implementation of social research. No individual-level data were used, and no data could be linked to any individual. The study has received approval from the Ethics Committee Medical University in Warsaw (no. AKBE/77/23).

## 3. Results

### 3.1. Characteristics of Studied Participants

The sociodemographic characteristics of the studied group are summarized in Table 1. It included a larger percentage of women and was represented by various age groups. Most of the individuals had completed secondary education and inhabited rural areas (Table 1). 

### 3.2. Attitudes toward Influenza Vaccination

One in three surveyed individuals (36.3%, 292/805) declared they strongly supported vaccination, while four in ten (41.2%, 332/805) expressed a moderate level of support. Moderate opponents were represented by one in six surveyed (15.8%, 127/805), while 6.7% (54/805) declared themselves as strong opponents of vaccination. Those strongly in favor of vaccination were statistically significantly more likely to be middle-aged and older (over 50) rather than younger people (*p* < 0.001). The rate of strong supporters of vaccination started to increase in individuals aged 35 and older, with the highest rate found in those aged 65 and older (Table 2).

### 3.3. The Influence of Medical and Personal Authority on Influenza Vaccination

A physician’s opinion on influenza vaccination would have very little influence for one in nine (11.0%, 89/805) respondents and relatively little influence for 5.7% (46/805). Nearly one in five (18.4%, 148/805) said a physician’s opinion would have neither little nor much influence. For 28.9% (233/805) of people, a physician’s recommendation to get vaccinated against influenza would be rather important, and for one in three people (35.9%, 289/805), it would be very important (*MD* = 4.00). The minor influence of a physician’s opinion on influenza vaccination was indicated significantly more often by middle-aged people between 35–49 years of age (*p* = 0.003). The most significant influence of a physician’s opinion on influenza vaccination was indicated significantly more often by older people 65 (*p* = 0.003) (Table 3).

The nurse’s opinion on the decision to vaccinate against influenza would have very little influence for 17.9% (144/805) of respondents and relatively little influence for 12.1% (97/805). One in three (33.4%, 269/805) of respondents reported that the nurse’s opinion would have neither little nor much influence. For one in five (26.7%, 215/805), a nurse’s recommendation to get vaccinated against influenza would be rather important, and for one in ten (9.9%, 79/805), it would be very important (*MD* = 3.00).

A pharmacist’s opinion on influenza vaccination would have very little influence for 18.3% (147/805) of respondents, rather little influence for 13.1% (106/805) of respondents, and one in three (34.0%, 274/805) declared that a pharmacist’s opinion would have neither little nor much influence. For one in five (26.8%, 215/805), a pharmacist’s recommendation to get vaccinated against influenza would be rather important, and for 7.8% (62/805), it would be very important (*MD* = 3.00). The statistically significant most minor influence of a pharmacist’s opinion on influenza vaccination was indicated more often by women than men (*p* = 0.011) and by those of middle age between 35–49 (*p* = 0.033). The statistically significant greatest influence of a pharmacist’s opinion on influenza vaccination was indicated more often by men than women (*p* = 0.011) (Table 3). 

A family member/friend’s opinion on influenza vaccination would have a minimal influence for 16.9% (136/805) of respondents and a rather small influence for 18.1% (146/805). One in three (33.6%, 270/805) reported that a family member/friend’s opinion would have neither a small nor an important influence. For one in five (24.2%, 195/805), a family member/friend’s recommendation to get vaccinated against influenza would be rather important, and for 7.2% (58/805), it would be very important (*MD* = 3.00). The minor influence of a family member’s/friend’s opinion on getting vaccinated against influenza was statistically significantly more often indicated by those of middle age between 35–49 (*p* = 0.043). The highest influence of a family member/friend’s opinion on getting vaccinated against influenza was indicated by young people between 25–34 years old (*p* = 0.043) (Table 3).

### 3.4. Attitudes toward Vaccination Compared to Levels of Trust in Particular Authority Groups

Among those who are strong proponents of vaccination to a high degree, the opinion of a physician in recommending influenza vaccination would be considered by six in ten (59.7%) respondents (*p* < 0.001) (Table 4 and Figure 1); the opinion of a nurse would be considered by two-thirds (67.1%) of respondents (*p* < 0.001) (Table 4 and Figure 1); a pharmacist’s opinion would be relied upon by one in two (51.6%) respondents (*p* < 0.001) (Table 4 and Figure 1). A family member/friend’s opinion would be relied upon by four in ten (42.1%) people (*p* < 0.001) (Table 4 and Figure 1).

Among those who were strong opponents of vaccination, the opinion of a physician was strongly disregarded by three in ten (29.5%) respondents (*p* < 0.001) (Table 4 and Figure 1); the opinion of a nurse in recommending influenza vaccination was strongly disregarded by 18.1% of respondents (*p* < 0.001) (Table 4 and Figure 1); the opinion of a pharmacist was strongly disregarded by 18.4% of respondents (*p* < 0.001) (Table 4 and Figure 1); and the opinion of a family member/friend was strongly disregarded by one in five (20.4%) people (*p* < 0.001) (Table 4 and Figure 1).

## 4. Discussion

The present study examined what effect the opinion of socially recognized medical authorities (physicians, nurses, pharmacists) and subjectively recognized personal authorities (friends/family members) has on people’s decision to be vaccinated against influenza in Poland. We believe that these observations are important for those who wish to improve influenza vaccination uptake, which remains very low among Poles [35].

There was a significant relationship between the age of the surveyed individuals and their attitude towards vaccination. The percentage of those who declared themselves as strong vaccine supporters increased with age, with a negligible share of strong opponents seen in individuals aged over 65 years. These findings are in line with other observations. In the case of influenza vaccination, the average vaccination rate in Poland among the elderly is much higher than that for the entire Polish population [38]. 

A 2021 U.S. study among outpatients indicated that general vaccine safety perception is statistically significantly associated with age, with older people over 50 almost twice as likely as those under 30 to believe that vaccination is safe [40]. A Polish study comparing reasons for not taking the flu vaccine from the point of view of physicians and their patients indicated that one of the important elements influencing patients’ decision to be vaccinated against influenza was older age and the common age-related experience of chronic illness [41]. Moreover, in a Japanese study, influenza vaccination was also statistically significantly associated with higher age [42]. Similar trends were also seen in the case of other vaccines against respiratory diseases, e.g., COVID-19 [43] and pneumococcus infection [44].

Individuals who declared themselves to be strong vaccine supporters were significantly more likely to rely on the opinion of medical authorities (i.e., physicians, nurses, and pharmacists) for influenza vaccination than those who declared they were strong vaccine opponents. Several studies confirm the effect of a physician’s recommendation on influenza vaccination. A German study of pregnant women indicated that the vaccination rate among those who received a recommendation from their gynecologist or general practitioner was significantly higher. Moreover, only 3.3% of women who did not receive a recommendation to vaccinate against influenza from their physician eventually decided to receive a vaccine [45]. A Polish study among teachers indicated that a physician’s recommendation to get vaccinated against influenza and the experience of a vaccinated family member were the main influences on increased vaccination rates [46]. Moreover, among Turkish patients, the predominant belief was that a physician’s recommendation for influenza vaccination would strongly influence their decision to accept it [47]. In a South African study of diabetics, a physician’s recommendation was particularly influential among those with previous experience with influenza vaccination [48]. Although it may appear intuitive that healthcare professionals represent a driving force for vaccination among the general public, one should note that our study was conducted in late 2022, following the high activity of anti-vaccine movements during the COVID-19 pandemic, which contributed to a spread of misinformation about vaccination on an unprecedented scale [49,50,51,52,53]. This phenomenon could potentially affect the overall trust in medical professionals as an authority regarding decisions on various vaccinations. The devastating effect of misinformation was demonstrated in the study of unvaccinated patients hospitalized with severe COVID-19. For one-third of patients, the personal experience with the disease did not change their primary refusal of vaccination, driven by personal beliefs and discouraged by online information, friends, and relatives [34]. Moreover, during the COVID-19 pandemic, the percentage of parents who refuse immunization for their children continued to increase in Poland [54,55].

Our study indicates that physicians have the most significant influence on patients’ attitudes toward vaccination. No recent representative studies show what percentage of physicians in Poland are vaccinated against influenza, although some data indicate that it is far from satisfactory, e.g., in the 2016/2017 epidemic season, it amounted only to 32.2%, with the majority of them being pediatricians and general practitioners [54]. These observations and our study’s results imply that the promotion of influenza vaccinations in Poland should first focus on increasing the vaccination rate among physicians, who will subsequently influence their patients and the general public. However, other healthcare professionals can also impact influenza vaccination rates, as also highlighted by a meta-analysis conducted in data from Singapore [56]. Moreover, a study conducted in a U.S. hospital observed the effect of pharmacists’ recommendations on increasing vaccination rates. Patients who were eligible for influenza and pneumococcal vaccination, but declined, were personally educated by pharmacists in the pharmacy, after which they were offered vaccination again. As a result, one in four people agreed to be vaccinated against influenza [57]. As calculated in Canada, including pharmacists in consultation services among seniors for influenza vaccination is cost-effective and improves vaccination rates in this group [58]. All in all, future campaigns aimed at increasing the rate of influenza vaccination in Poland should integrate various groups of healthcare workers, not only physicians, and should include pharmacists, as they are likely to be role models for decisions undertaken by the general public.

In addition, our study demonstrated that the age of respondents played a significant role in the influence of the physician’s opinion and that of a friend/family member on influenza vaccination. Those who declared that a physician’s opinion on this vaccination mattered greatly were mostly the oldest people over 65. A Tunisian study among the elderly indicated that a physician’s recommendation was the main reason that led to flu vaccination [59]. These observations can be explained by more frequent interactions with healthcare workers, including physicians, in the group of older individuals. In contrast, younger adults interact less often with a healthcare professional and less frequently suffer from direct consequences of influenza infection [60]. Therefore, the campaigns promoting influenza vaccination should not only highlight the benefits of influenza vaccination in the younger groups but also maximize the role of different healthcare professionals in communicating these benefits.

In the case of personal authority, i.e., a friend/family member, the difference between proponents and opponents of vaccination in terms of the influence of this authority on the decision to vaccinate against influenza was no longer as pronounced, although it remained statistically significant. Nevertheless, a Japanese study showed that the influence of a family member on the issue of influenza vaccination could be stronger than a physician’s opinion [61]. Subjective norms created by people based on the opinion of those closest to them in the social environment can strongly influence the decision to vaccinate against influenza. While compliance with descriptive, objective norms like “most people get vaccinated” influences beliefs about the value of vaccination, encouraging friends and loved ones of the person who should be vaccinated to talk about flu vaccination seems to translate into action [62]. In our study, respondents who reported a very strong influence of a friend/family member on their decision to be vaccinated against influenza were significantly more likely to be aged 25–34 years. Qualitative research among young Polish immigrants in Scotland has indicated that Polish people’s health decisions, including vaccination, are most influenced by friends and acquaintances [63].

### Limitations and Strengths of the Study

The present study has several limitations. First, during the implementation of the survey, work was underway in the Polish parliament to increase the rights of pharmacists in qualifying, prescribing, and vaccinating patients against influenza. This may have influenced respondents’ answers during the survey implementation. Although, at the same time, the value of the presented results in the context of these legal changes and the public discussion triggered by this fact is much greater. Secondly, it might have been worthwhile to include in the study the influence of authorities on the decision to vaccinate against influenza celebrities and social media influencers, who often speak out about vaccination despite their lack of knowledge. However, in the research assumptions, we wanted to include only those authorities with whom the communication and recommendation are direct, in personal contact, and not mediated by a medium such as a social network or television. The great value of the survey is that it has a high degree of representativeness for Polish society and was carried out when the epidemic season began. Hence, influenza vaccination was a close topic of consideration and decision. However, one should note that not all declarations expressed by our study participants may translate into actual decisions to receive an influenza vaccine. Nevertheless, this research aimed not to assess vaccine intake rates but to explore sociodemographic factors that may influence the acceptance or refusal of influenza vaccination in Poland.

## 5. Conclusions

The present research confirms that the people who enjoy the most considerable authority among respondents when it comes to getting vaccinated against influenza are physicians, followed by other medical professionals, i.e., nurses and pharmacists. As pharmacists’ authority to qualify for and administer vaccinations has increased over the past few years and will also expand shortly to include the ability to write prescriptions for flu vaccines, it was essential to include this group in the cross-sectional survey. The study indicates that confidence in pharmacists regarding influenza vaccination, especially among vaccination advocates, is high. Considering the results of the presented study and the discussion based on it with the work of other authors on this topic, it is worth noting that in the process of educating the public about influenza vaccination, the role of physicians, nurses, and pharmacists as medical authorities should be strengthened. However, the role of personal authorities, i.e., friends and family members, who influence the formation of people’s subjective social norms, should not be forgotten.

## Figures and Tables

**Figure 1 vaccines-11-00994-f001:**
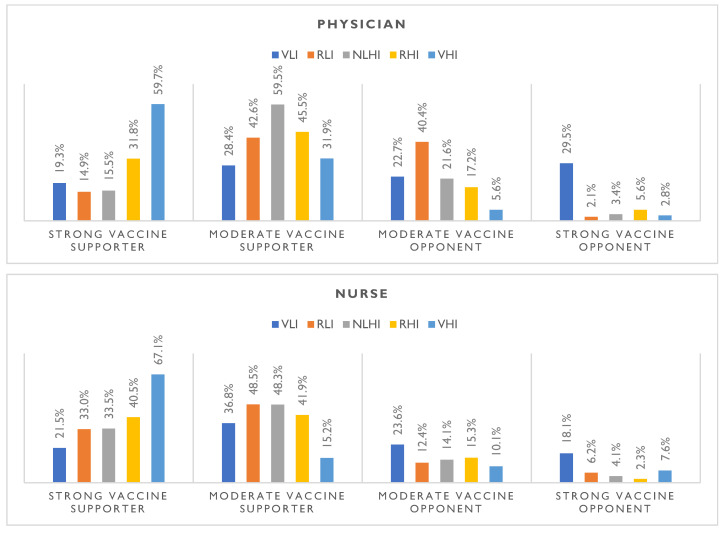
Summary of the intensity of the level of influence of the particular authority group’s opinion on the decision to vaccinate against influenza and the declared attitude of respondents towards vaccination (*N* = 805). Note. VLI—very low influence, RLI—rather low influence, NLHI—neither low nor high influence, RHI—rather high influence, VHI—very high influence.

**Table 1 vaccines-11-00994-t001:** Sociodemographic characteristics of the study population (*N* = 805).

	*n*	%
Sex		
Female	428	53.1
Male	377	46.9
Age		
18–24	100	12.4
25–34	153	19.0
35–49	195	24.3
50–64	231	28.7
65 and more	126	15.7
Education		
Primary	29	3.6
Vocational	59	7.3
Secondary	411	51.1
Tertiary	306	38.0
Place of living		
Rural area	323	40.1
City < 20 k	94	11.7
City 20–99 k	154	19.1
City 100–199 k	66	8.2
City 200–499 k	73	9.0
City > 500 k	95	11.8

**Table 2 vaccines-11-00994-t002:** Respondents’ attitudes toward influenza vaccination by demographic characteristics (*N* = 805).

Demographic Categories	Vaccination Supportes		Vaccination Opponents		*p*-Value
	Strong	Moderate	Strong	Moderate	
(*n*/%)					
Sex					
Female	146/34.2	187/43.8	69/16.2	25/5.9	0.309
Male	146/38.6	145/38.4	58/15.3	29/7.7	
Age					
18–24	29/28.7	39/38.6	25/24.8	8/7.9	<0.001
25–34	40/26.1	77/50.3	23/15.0	13/8.5	
35–49	61/31.3	78/40.0	40/20.5	16/8.2	
50–64	96/41.6	91/39.4	27/11.7	17/7.4	
65 and more	67/53.2	47/37.3	11/8.7	1/0.8	
Education					
Primary	10/34.5	12/41.4	6/20.7	1/3.4	0.298
Vocational	24/40.7	21/35.6	9/15.3	5/8.5	
Secondary	135/32.8	171/41.5	75/18.2	31/7.5	
Tertiary	124/40.5	129/42.2	36/11.8	17/5.6	
Place of living					
Country	111/34.4	132/40.9	59/18.3	21/6.5	0.190
City < 20 k	35/37.2	36/38.3	14/14.9	9/9.6	
City 20–99 k	55/35.9	70/45.8	22/14.4	6/3.9	
City 100–199 k	23/34.2	28/41.8	15/22.4	1/1.5	
City 200–499 k	28/38.9	30/41.7	5/6.9	9/12.5	
City > 500 k	39/41.1	36/37.9	12/12.6	8/8.4	

**Table 3 vaccines-11-00994-t003:** Relationship between attitudes toward vaccination and demographic characteristics of respondents (*N* = 805).

	Influence					Median	Percentiles			*p*-Value
Sex and Age of Respondents	Very Low	Rather Low	Neither Low nor High	Rather High	Very High		25th	50th	75th	
Physician										
Sex										
Female	58/13.6	24/5.6	76/17.8	118/27.6	151/35.4	4.00	3.00	4.00	5.00	0.151
Male	30/7.9	22/5.8	73/19.3	115/30.4	138/36.5	4.00	3.00	4.00	5.00	
Age										
18–24	8/8.0	8/8.0	17/17.0	30/30.0	37/37.0	4.00	3.00	4.00	5.00	0.003
25–34	14/9.1	10/6.5	38/24.7	49/31.8	43/27.9	4.00	3.00	4.00	5.00	
35–49	31/16.0	17/8.8	31/16.0	56/28.9	59/30.4	4.00	2.62	4.00	5.00	
50–64	28/12.1	9/3.9	41/17.7	66/28.6	87/37.7	4.00	3.00	4.00	5.00	
65 and more	7/5.6	2/1.6	21/16.8	32/25.6	63/50.4	4.88	4.00	4.88	5.00	
Nurse										
Sex										
Female	84/19.6	55/12.9	145/33.9	108/25.2	36/8.4	3.00	2.00	3.00	4.00	0.314
Male	61/16.1	42/11.1	124/32.8	107/28.3	44/11.6	3.00	2.00	3.00	4.00	
Age										
18–24	19/19.0	16/16.0	31/31.0	24/24.0	10/10.0	3.00	2.00	3.00	4.00	0.272
25–34	21/13.8	20/13.2	53/34.9	39/25.7	19/12.5	3.00	2.00	3.00	4.00	
35–49	45/23.1	22/11.3	55/28.2	54/27.7	19/9.7	3.00	2.00	3.00	4.00	
50–64	48/20.6	27/11.6	83/35.6	57/24.5	18/7.7	3.00	2.00	3.00	4.00	
65 and more	13/10.3	12/9.5	47/37.3	41/32.5	13/10.3	3.00	3.00	3.00	4.00	
Pharmacist										
Sex										
Female	95/22.2	63/14.7	133/31.1	107/25.0	30/7.0	3.00	2.00	3.00	4.00	0.011
Male	53/14.0	43/11.4	141/37.3	109/28.8	32/8.5	3.00	2.00	3.00	4.00	
Age										
18–24	18/18.0	14/14.0	37/37.0	25/25.0	6/6.0	3.00	2.00	3.00	4.00	0.033
25–34	21/13.8	25/16.4	58/38.2	32/21.1	16/10.5	3.00	2.00	3.00	4.00	
35–49	47/24.1	27/13.8	59/30.3	47/24.1	15/7.7	3.00	2.00	3.00	4.00	
50–64	49/21.2	30/13.0	74/32.0	63/27.3	15/6.5	3.00	2.00	3.00	4.00	
65 and more	13/10.2	10/7.9	46/36.2	48/37.8	10/7.9	3.00	3.00	3.00	4.00	
Friend/Family Member										
Sex										
Female	81/19.0	79/18.5	137/32.1	97/22.7	33/7.7	3.00	2.00	3.00	4.00	0.378
Male	55/14.6	66/17.5	133/35.3	98/26.0	25/6.6	3.00	2.00	3.00	4.00	
Age										
18–24	12/12.0	21/21.0	32/32/0	27/27.0	7/7.0	3.00	2.00	3.00	4.00	0.043
25–34	20/13.0	28/18.2	53/34.4	32/20.8	21/13.6	3.00	2.00	3.00	4.00	
35–49	43/22.1	34/17.4	59/30.3	46/23.6	13/6.7	3.00	2.00	3.00	4.00	
50–64	48/20.8	38/16.5	76/32.9	57/24.7	12/5.2	3.00	2.00	3.00	4.00	
65 and more	14/11.1	24/19.0	50/39.7	33/26.2	5/4.0	3.00	2.00	3.00	4.00	

**Table 4 vaccines-11-00994-t004:** The relationship between attitudes toward vaccination and the level of trust in influenza vaccination recommendations to individual authorities.

Attitude towards Vaccination	Median	Percentiles			*p*-Value
		25th	50th	75th	
	Physician				
Strong vaccine supporter	5.00	4.00	5.00	5.00	<0.001
Moderate vaccine supporter	4.00	3.00	4.00	5.00	
Moderate vaccine opponent	3.00	2.00	3.00	4.00	
Strong vaccine opponent	2.44	1.00	2.44	4.00	
	Nurse				
Strong vaccine supporter	3.00	3.00	3.00	4.00	<0.001
Moderate vaccine supporter	3.00	2.00	3.00	4.00	
Moderate vaccine opponent	3.00	1.00	3.00	4.00	
Strong vaccine opponent	2.00	1.00	2.00	3.00	
	Pharmacist				
Strong vaccine supporter	3.00	3.00	3.00	4.00	<0.001
Moderate vaccine supporter	3.00	2.00	3.00	4.00	
Moderate vaccine opponent	3.00	1.00	3.00	4.00	
Strong vaccine opponent	1.64	1.00	1.64	3.41	
	Friend/Family Member				
Strong vaccine supporter	3.00	2.00	3.00	4.00	<0.001
Moderate vaccine supporter	3.00	2.00	3.00	4.00	
Moderate vaccine opponent	3.00	2.00	3.00	4.00	
Strong vaccine opponent	1.00	1.00	1.00	4.00	

## Data Availability

The data presented in this study are available upon request from the corresponding author.

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
