# Peer review of "The Influence of Recommendation of Medical and Non-Medical Authorities on the Decision to Vaccinate against Influenza from a Social Vaccinology Perspective: Cross-Sectional, Representative Study of Polish Society"

_vaccines, 2023, doi:10.3390/vaccines11050994_

Round 1

Reviewer 1 Report

I have carefully read and reviewed the paper ‘The Influence of Recommendation of Medical and Non-medical Authorities on the Decision to Vaccinate Against Influenza from a Social Vaccinology Perspective: Cross-Sectional, Representative Study of Polish Society’ submitted by Tomasz Sobierajski, Piotr Rzymski and Monika Wanke-Rytt. As I understand it, the aim of this work is to to know the reasons for such a  low level of vaccination and to assess the influence of medical and social authorities on the decision  to vaccinate against influenza from the perspective of social vaccinology. The author believes that the need to strengthen the authority of physicians and pharmacists regarding influenza vaccination, also in the case of pharmacists in terms of changing the law to allow them to qualify for influenza vaccination.

Although this is a well-researched paper, I think it cannot be recommended for publication before revision. My reasons for this are listed below.

Major:

1. Some opinions need to be supported by references or data. The authors argue that some authorities are culturally universally recognized (e.g., doctors), while others are personally felt (e.g., friends). It is understandable that a doctor has authority during his or her work, but if friends also have authority, then some people will raise objections. If this view is not true, it will affect the scientific nature of the research design. (Lines 99-101)

2. The authors propose that individual authorities, not recognized by the public, may also influence one's actions, including those related to preventive health care, such as vaccination. These include friends or family members who can be classified as deontic authorities by their function. In the paper (Reference 22) cited by the author, it is proposed that parents and friends who figure as adolescents' most salient interpersonal relationships, without raising the question of authority. The paper (Reference 22) only studies young people and does not include other age groups. Teenagers (including minors) are easily influenced by parents and friends. It does not mean that all people are easily influenced by parents and friends, and if so, the influence varies. (Lines 111-114)

3. The authors believe that those strongly in favour of vaccination were statistically significantly more likely to be middle-aged and older (over 50) than younger people. However, among the surveyed population, there were 357 people aged 50 and above, accounting for 44.45% of the total population. Therefore, this has a direct impact on the research results, leading to deviations in the author's understanding. (Lines 212-216)

4. The authors put forward that the people who enjoy the most considerable authority among respondents when it comes to getting vaccinated against influenza are physicians, followed by other medical professionals, i.e., nurses and pharmacists. (Lines 360--362) The conclusion of this study is consistent with the public's daily cognition. Even without this research, such a conclusion can still be drawn, therefore, the author needs to further explain the value of this study.

Minor:

5. The format of references is not standardized, which needs to be revised.

Author Response

Reviewer 1

I have carefully read and reviewed the paper 'The Influence of Recommendation of Medical and Non-medical Authorities on the Decision to Vaccinate Against Influenza from a Social Vaccinology Perspective: Cross-Sectional, Representative Study of Polish Society' submitted by Tomasz Sobierajski, Piotr Rzymski and Monika Wanke-Rytt. As I understand it, the aim of this work is to to know the reasons for such a  low level of vaccination and to assess the influence of medical and social authorities on the decision  to vaccinate against influenza from the perspective of social vaccinology. The author believes that the need to strengthen the authority of physicians and pharmacists regarding influenza vaccination, also in the case of pharmacists in terms of changing the law to allow them to qualify for influenza vaccination.

Although this is a well-researched paper, I think it cannot be recommended for publication before revision. My reasons for this are listed below.

Author's response: We would like to thank the Reviewer for throughout the review of our manuscript, positive feedback towards it, and critical comments, which we followed when revising the manuscript.  

  1. Some opinions need to be supported by references or data. The authors argue that some authorities are culturally universally recognized (e.g., doctors), while others are personally felt (e.g., friends). It is understandable that a doctor has authority during his or her work, but if friends also have authority, then some people will raise objections. If this view is not true, it will affect the scientific nature of the research design. (Lines 99-101)

Author's response: We have examined the manuscript to detect parts without references to support the statements and added citation to various works when necessary. The following references were added:

  • Hughes, M.M.; Reed, C.; Flannery, B.; Garg, S.; Singleton, J.A.; Fry, A.M.; Rolfes, M.A. Projected Population Benefit of Increased Effectiveness and Coverage of Influenza Vaccination on Influenza Burden in the United States. Clin. Infect. Dis. 2020, 70, 2496–2502.
  • Foppa, I.M.; Cheng, P.-Y.; Reynolds, S.B.; Shay, D.K.; Carias, C.; Bresee, J.S.; Kim, I.K.; Gambhir, M.; Fry, A.M. Deaths Averted by Influenza Vaccination in the U.S. during the Seasons 2005/06 through 2013/14. Vaccine 2015, 33, 3003–3009.
  • CDC 2021–2022 Estimated Flu Illnesses, Medical Visits, Hospitalizations, and Deaths Prevented by Flu Vaccination Available online: https://www.cdc.gov/flu/about/burden-averted/2021-2022.htm (accessed on 7 May 2023).
  • Bloom, D.E.; Cadarette, D.; Ferranna, M. The Societal Value of Vaccination in the Age of COVID-19. J. Public Health 2021, 111, 1049–1054.
  • de Figueiredo, A.; Simas, C.; Larson, H.J. COVID-19 Vaccine Acceptance and Its Socio-Demographic and Emotional Determinants: A Multi-Country Cross-Sectional Study. Vaccine 2023, 41, 354–364.
  • Vet, R.; de Wit, J.B.; Das, E. Factors Associated with Hepatitis B Vaccination among Men Who Have Sex with Men: A Systematic Review of Published Research. Int. J. STD AIDS 2017, 28, 534–542.
  • Sheinfeld Gorin, S.N.; Glenn, B.A.; Perkins, R.B. The Human Papillomavirus (HPV) Vaccine and Cervical Cancer: Uptake and next Steps. Adv. Ther. 2011, 28, 615–639.
  • Prematunge, C.; Corace, K.; McCarthy, A.; Nair, R.C.; Pugsley, R.; Garber, G. Factors Influencing Pandemic Influenza Vaccination of Healthcare Workers--a Systematic Review. Vaccine 2012, 30, 4733–4743.
  • Sofaer, S.; Kreling, B.; Kenney, E.; Swift, E.K.; Dewart, T. Family Members and Friends Who Help Beneficiaries Make Health Decisions. Health Care Financ. Rev. 2001, 23, 105–121.
  • Watts, M.S. Physicians as Role Models in Society. West. J. Med. 1990, 152, 292.
  • Shen, A.K.; Browne, S.; Srivastava, T.; Michel, J.J.; Tan, A.S.L.; Kornides, M.L. Factors Influencing Parental and Individual COVID-19 Vaccine Decision Making in a Pediatric Network. Vaccines 2022, 10, 1277.
  • Zarębska-Michaluk, D.; Rzymski, P.; Moniuszko-Malinowska, A.; Brzdęk, M.; Martonik, D.; Rorat, M.; Wielgat, J.; Kłos, K.; Musierowicz, W.; Wasilewski, P.; et al. Does Hospitalization Change the Perception of COVID-19 Vaccines among Unvaccinated Patients? Vaccines 2022, 10, 476.
  • Quandelacy, T.M.; Viboud, C.; Charu, V.; Lipsitch, M.; Goldstein, E. Age- and Sex-Related Risk Factors for Influenza-Associated Mortality in the United States between 1997-2007. Am. J. Epidemiol. 2014, 179, 156–167.
  • van der Linden, S.; Roozenbeek, J.; Compton, J. Inoculating against Fake News about COVID-19. Psychol. 2020, 11, 566790.
  • Rzymski, P.; Borkowski, L.; Drąg, M.; Flisiak, R.; Jemielity, J.; Krajewski, J.; Mastalerz-Migas, A.; Matyja, A.; Pyrć, K.; Simon, K.; et al. The Strategies to Support the COVID-19 Vaccination with Evidence-Based Communication and Tackling Misinformation. Vaccines (Basel) 2021, 9, 109.
  • Nowak, B.M.; Miedziarek, C.; Pełczyński, S.; Rzymski, P. Misinformation, Fears and Adherence to Preventive Measures during the Early Phase of COVID-19 Pandemic: A Cross-Sectional Study in Poland. Int. J. Environ. Res. Public Health 2021, 18, 12266.
  • Skafle, I.; Nordahl-Hansen, A.; Quintana, D.S.; Wynn, R.; Gabarron, E. Misinformation about COVID-19 Vaccines on Social Media: Rapid Review. J. Med. Internet Res. 2022, 24, e37367.
  • Lee, S.K.; Sun, J.; Jang, S.; Connelly, S. Misinformation of COVID-19 Vaccines and Vaccine Hesitancy. Sci. Rep. 2022, 12, 13681.
  • Kraśnicka, J.; Krajewska-Kułak, E.; Klimaszewska, K.; Cybulski, M.; Guzowski, A.; Kowalewska, B.; Jankowiak, B.; Rolka, H.; Doroszkiewicz, H.; Kułak, W. Mandatory and Recommended Vaccinations in Poland in the Views of Parents. Vaccin. Immunother. 2018, 14, 2884–2893.
  • Narodowy Instytut Zdrowia Publicznego – Państwowy Zakład Higieny Jaka jest liczba uchyleń dotyczących szczepień obowiązkowych? Available online: https://szczepienia.pzh.gov.pl/faq/jaka-jest-liczba-uchylen-szczepien-obowiazkowych/ (accessed on 7 May 2023).
  1. The authors propose that individual authorities, not recognized by the public, may also influence one's actions, including those related to preventive health care, such as vaccination. These include friends or family members who can be classified as deontic authorities by their function. In the paper (Reference 22) cited by the author, it is proposed that parents and friends who figure as adolescents' most salient interpersonal relationships, without raising the question of authority. The paper (Reference 22) only studies young people and does not include other age groups. Teenagers (including minors) are easily influenced by parents and friends. It does not mean that all people are easily influenced by parents and friends, and if so, the influence varies. (Lines 111-114)

Author's response: Thank you for noting it. To balance the previously cited reference, we have added a reference to the study which examined the perception of vaccines among patients hospitalized with severe COVID-19 (mean age: 59 years), which also shows that friends and family play a significant role in such decisions, particularly in vaccine refusal (doi: 10.3390/vaccines10030476).

  1. The authors believe that those strongly in favour of vaccination were statistically significantly more likely to be middle-aged and older (over 50) than younger people. However, among the surveyed population, there were 357 people aged 50 and above, accounting for 44.45% of the total population. Therefore, this has a direct impact on the research results, leading to deviations in the author's understanding. (Lines 212-216)

Author's response: Thank you for this comment. However, please note that in Table 2, the results are also presented as %, and one can clearly see that the percentage of strong supporters of vaccination increases with age, while the percentage of strong opponents decreases. This is why we have used five different age groups and compared them. To strengthen the meaning of the description of the results, we have added the following line: "The rate of strong supporters of vaccination started to increase in individuals aged 35 and older, with the highest rate found in those aged 65 and older (Table 2)".

  1. The authors put forward that the people who enjoy the most considerable authority among respondents when it comes to getting vaccinated against influenza are physicians, followed by other medical professionals, i.e., nurses and pharmacists. (Lines 360--362) The conclusion of this study is consistent with the public's daily cognition. Even without this research, such a conclusion can still be drawn, therefore, the author needs to further explain the value of this study.

Author's response: Thank you for this comment. We have addressed it within the revised Discussion, indicating that the role of physicians as role models for the general public regarding vaccination could be influenced by the COVID-19 pandemic, the spread of misinformation, high activity of anti-vaccine groups. Moreover, we have put this issue in the context of low vaccination rates among Polish physicians, indicating that increasing interest in influenza vaccination in Poland would first require increasing the vaccination rate among physicians, who subsequently would inspire the general public. Several lines discussing it, and referencing associated data, were added.

  1. The format of references is not standardized, which needs to be revised.

Author's response: The professional citation manager software was employed in the revised manuscript to order and format the references.

Reviewer 2 Report

This study tried to figure out the influence of recommendation of medical and non-medical authorities on the decision to influenza vaccination. It’s an interesting topic, while there is some major concerns need to be addressed.

As this is a representative survey, but the author did not test the representativeness of the 805 selected respondents to the target population. In addition, they did not report the response rate of the survey. Did you get back all of the delivered questionnaire? It seems impossible.

 Have you validated the vaccination attitude? How to confirmed their vaccination status? There is no report in the manuscript. 

 The statistical analysis is weak. And what is MD in the table 3 and table 4?

Minor comments

There is some errors in the table2 or in the text description. For instance, Moderate support persons: in table 2 says 146+146=292, while in the text 293. For the moderate opponents; 58+69=127, while in the text is 126. There are discrepancies, please double check all of the numbers.

Author Response

Reviewer 2

This study tried to figure out the influence of recommendation of medical and non-medical authorities on the decision to influenza vaccination. It's an interesting topic, while there is some major concerns need to be addressed.

Author's response: Thank you for reviewing our manuscript. Please find the response to your comments below.

As this is a representative survey, but the author did not test the representativeness of the 805 selected respondents to the target population. In addition, they did not report the response rate of the survey. Did you get back all of the delivered questionnaire? It seems impossible.

Author’s response: The survey was conducted by a professional research company, which used its panel to obtain a representative sample group for the population. Creating panels that include randomly selected adults is a method that all leading research companies worldwide use. The panelists, who are also selected by stratification strata to increase representativeness, as described in the Methodology, participate in the survey via the web. Panelists who do not have internet access at home - especially older people - are provided with a tablet. Nowadays, in virtually no developed country, surveys are no longer conducted based on a basic sampling frame, i.e., statistical data of all respondents. The exceptions are surveys conducted by the government. It is due, firstly, to the high cost of such a survey and the significant number of survey returns. Secondly, regulations related to data protection do not allow private individuals to use these sampling frames. For this reason, the only way to obtain population-representative responses from the general public is for survey companies to set up survey panels with selected panelists who are representative of the public. As a result, the percentage of refusals to respond is also negligible.

Have you validated the vaccination attitude? How to confirmed their vaccination status? There is no report in the manuscript. 

Author's response: This is a good comment. Obviously, our study was not designed to verify the responses other than the actual decisions. However, we agree that this point needs to be taken into account when discussing the study's limitations. Therefore, we have added the following line in the revised manuscript: "However, one should note that not all declarations expressed by our study participants may translate into the actual decisions to receive an influenza vaccine. Nevertheless, this research aimed not to assess vaccine intake rates but to explore sociodemographic factors that may influence the acceptance or refusal of influenza vaccination in Poland."

 The statistical analysis is weak. And what is MD in the table 3 and table 4?

Author's response: We have employed standard statistical methods to assess the differences between the groups, and since the data was expressed in Likert-type scales, a non-parametric Kruskal-Wallis ANOVA was used, as explained in subsection 2.4. "Statistical analysis". We have changed “MD” to "Median". The results were reported as median because Likert-type scales are ordinal ones; https://www.st-andrews.ac.uk/media/ceed/students/mathssupport/Likert.pdf

There is some errors in the table2 or in the text description. For instance, Moderate support persons: in table 2 says 146+146=292, while in the text 293. For the moderate opponents; 58+69=127, while in the text is 126. There are discrepancies, please double check all of the numbers.

Author’s response:  Thank you for this observation. There was indeed an error in the table description. We have corrected it.

Reviewer 3 Report

This is a well written study with appropriate Tables and Figures. My only suggestion would be to ask if you are able to construct 3-dimensional diagrams involving multiple parameters. This would add to the novelty of an already good paper.

The following comments do not in any way detract from the paper but may be worth mentioning - completely at your own discretion.

I appreciate that vaccination per se is a vastly important health intervention, however, my own recent research regarding both influenza and COVID-19 vaccination is that the vaccines can deliver totally unexpected outcomes, i.e. higher all-cause mortality, when assessed against all-cause mortality rather than just vaccine effectiveness. In this respect I would recommend a recent review Implications of Non-Specific Effects for Testing, Approving, and Regulating Vaccines - PubMed (nih.gov).

The issue here is that trust is based upon available knowledge, and that trust in respected figures (within the limits of available knowledge) needs to be maintained should conflicting medical information become available.

In this respect, my conclusion is that as an individual while I can wholeheartedly recommend the vast majority of vaccines the available new evidence suggests that I would be unable to recommend either way regarding influenza or COVID-19 vaccination. A person-based approach is required but we currently lack the knowledge as to which are the important parameters, i.e., genetic polymorphs, small noncoding RNA profiles, specific age profiles for morbidity/mortality of each clade/variant, etc.

You are free to make no comments regarding these issues as they do not directly impact on the specifics of your paper.

Author Response

Reviewer 3

This is a well written study with appropriate Tables and Figures. My only suggestion would be to ask if you are able to construct 3-dimensional diagrams involving multiple parameters. This would add to the novelty of an already good paper. The following comments do not in any way detract from the paper but may be worth mentioning - completely at your own discretion.

Author's response: Thank you for reading and reviewing our work and expressing positive feedback toward it. We also thank you for the additional discussion, which we have considered and addressed below.

I appreciate that vaccination per se is a vastly important health intervention, however, my own recent research regarding both influenza and COVID-19 vaccination is that the vaccines can deliver totally unexpected outcomes, i.e. higher all-cause mortality, when assessed against all-cause mortality rather than just vaccine effectiveness. In this respect I would recommend a recent review Implications of Non-Specific Effects for Testing, Approving, and Regulating Vaccines - PubMed (nih.gov). The issue here is that trust is based upon available knowledge, and that trust in respected figures (within the limits of available knowledge) needs to be maintained should conflicting medical information become available. In this respect, my conclusion is that as an individual while I can wholeheartedly recommend the vast majority of vaccines the available new evidence suggests that I would be unable to recommend either way regarding influenza or COVID-19 vaccination. A person-based approach is required but we currently lack the knowledge as to which are the important parameters, i.e., genetic polymorphs, small noncoding RNA profiles, specific age profiles for morbidity/mortality of each clade/variant, etc. You are free to make no comments regarding these issues as they do not directly impact on the specifics of your paper.

Author's response: Thank you for this comment. Please note that clinical trials, even in phase III, are not designed to detect very rare adverse events or events which are highly specific to a particular group of individuals within the population. This is not due to anyone's bad intentions, it's just beyond the design of these pivotal studies to authorize any medical intervention. It is also virtually impossible to include all the possible parameters during such investigations. A person-based approach is obviously highly valued in medical interventions, including vaccinations, but it is, at times, impossible to achieve, and if achieving it would be a necessary step to authorize any medical interventions, we are afraid that no intervention could be authorized. We believe that these are all very interesting issues, although their scope falls outside of our study.

Nevertheless, we wish to comment on COVID-19 and influenza vaccinations. Mathematical modeling shows that COVID-19 vaccination averted 19.8 million lives in 2021 alone (doi: 10.1016/S1473-3099(22)00320-6); meaning (assuming the number of vaccinated individuals in 2021) that it took 227 to be vaccinated to save a single life. Although this data lack a person-based approach, it still shows that "global approach" is valid and brings significant benefits (please note that the referenced figures relate to deaths only, not to averted hospitalization, medical visits, post-COVID syndrome cases, and others, including economic parameters).

It is also estimated that during the 2021-2022 season, influenza vaccination prevented only in US, 1.8 million flu-related illnesses, 1,000,000 medical visits, 22,000 hospitalizations, and averted 1,000 deaths (https://www.cdc.gov/flu/about/burden-averted/2021-2022.htm). As long as the effectiveness of current influenza vaccines may be far from optimal, these figures clearly show the direct and indirect (economic, sociologic) benefits of influenza vaccination. In the revised manuscript, we have referenced some data showing the benefits of such vaccination, specifically:

  • Hughes, M.M.; Reed, C.; Flannery, B.; Garg, S.; Singleton, J.A.; Fry, A.M.; Rolfes, M.A. Projected Population Benefit of Increased Effectiveness and Coverage of Influenza Vaccination on Influenza Burden in the United States. Clin. Infect. Dis. 2020, 70, 2496–2502.
  • Foppa, I.M.; Cheng, P.-Y.; Reynolds, S.B.; Shay, D.K.; Carias, C.; Bresee, J.S.; Kim, I.K.; Gambhir, M.; Fry, A.M. Deaths Averted by Influenza Vaccination in the U.S. during the Seasons 2005/06 through 2013/14. Vaccine 2015, 33, 3003–3009.
  • CDC 2021–2022 Estimated Flu Illnesses, Medical Visits, Hospitalizations, and Deaths Prevented by Flu Vaccination Available online: https://www.cdc.gov/flu/about/burden-averted/2021-2022.htm (accessed on 7 May 2023).

Reviewer 4 Report

the introduction is very long and often out of the scope of the paper. Please, focus on the main aspects.

The questionnaire used should be added as supplementary material since it was developed ad hoc for the purpose of the current study.

variables are not described. As for instance how did you define vaccine supporters or opponents? At the same time categorization method is not explained. 

In the discussion, authors listed several similar studies, however, they look not well integrated in a flow. Please, revise focusing firstly on your mainly results, than comparing them with the literature and evaluating internal consistency of your results, and lastly discuss public health impact of your results.

moderate english revision should be provided

Author Response

Reviewer 4

General author's remark: Thank you for reviewing our manuscript and providing additional comments. We responded to each of them below.

the introduction is very long and often out of the scope of the paper. Please, focus on the main aspects.

Author's response: Thank you for this comment. Please note that length our Introduction, after revisions, is a little bit over 2 pages, while 2 pages appear to be within the standard length of this section. We believe that in the present form, the Introduction allows to build a background in detail, meet the biomedical model with the holistic model, and set it in the proper context (after all, the study was conducted in late 2022, while the COVID-19 pandemic had a profound effect on the perception of vaccines, not only COVID-19 vaccines). We believe that providing some sociological perspective in such a paper is necessary and may be of interest to some additional readers of the journal. Thus, we use the Introduction to make the paper interesting to a broad audience.

Please also kindly note that Vaccines have recently introduced a new rule according to which the manuscript reporting original data should have a minimum word count of 4,000 words. We believe that the intention was to have a background properly built and the results well-discussed, and we followed this guideline when preparing our manuscript. https://www.mdpi.com/journal/vaccines/instructions

The questionnaire used should be added as supplementary material since it was developed ad hoc for the purpose of the current study.

Author’s response: The questionnaire translated from polish to English was insertef in Supplementary Materials at the end of the manuscript. Relevant information has been added in the subsection The Questionnaire Lines 247-248.

variables are not described. As for instance how did you define vaccine supporters or opponents? At the same time categorization method is not explained. 

Author’s response: Respondents decided which group they would fall into in line with the survey design. They could choose between four groups arranged on a scale: strong vaccine opponent, moderate vaccine opponent, moderate vaccine supporter, strong vaccine supporter. We have added an explanation to this in the methodology. Lines 234-239

In the discussion, authors listed several similar studies, however, they look not well integrated in a flow. Please, revise focusing firstly on your mainly results, than comparing them with the literature and evaluating internal consistency of your results, and lastly discuss public health impact of your results.

Author’s response: Thank you for this remark. We have revised our Discussion to expand it when discussing various aspects, but also to integrate other research with ours in outlining specific conclusions or recommendations. Additional works were also referenced for this purpose, including:

  • Zarębska-Michaluk, D.; Rzymski, P.; Moniuszko-Malinowska, A.; Brzdęk, M.; Martonik, D.; Rorat, M.; Wielgat, J.; Kłos, K.; Musierowicz, W.; Wasilewski, P.; et al. Does Hospitalization Change the Perception of COVID-19 Vaccines among Unvaccinated Patients? Vaccines 2022, 10, 476.
  • Quandelacy, T.M.; Viboud, C.; Charu, V.; Lipsitch, M.; Goldstein, E. Age- and Sex-Related Risk Factors for Influenza-Associated Mortality in the United States between 1997-2007. Am. J. Epidemiol. 2014, 179, 156–167.
  • van der Linden, S.; Roozenbeek, J.; Compton, J. Inoculating against Fake News about COVID-19. Psychol. 2020, 11, 566790.
  • Rzymski, P.; Borkowski, L.; Drąg, M.; Flisiak, R.; Jemielity, J.; Krajewski, J.; Mastalerz-Migas, A.; Matyja, A.; Pyrć, K.; Simon, K.; et al. The Strategies to Support the COVID-19 Vaccination with Evidence-Based Communication and Tackling Misinformation. Vaccines (Basel) 2021, 9, 109.
  • Nowak, B.M.; Miedziarek, C.; Pełczyński, S.; Rzymski, P. Misinformation, Fears and Adherence to Preventive Measures during the Early Phase of COVID-19 Pandemic: A Cross-Sectional Study in Poland. Int. J. Environ. Res. Public Health 2021, 18, 12266.
  • Skafle, I.; Nordahl-Hansen, A.; Quintana, D.S.; Wynn, R.; Gabarron, E. Misinformation about COVID-19 Vaccines on Social Media: Rapid Review. J. Med. Internet Res. 2022, 24, e37367.
  • Lee, S.K.; Sun, J.; Jang, S.; Connelly, S. Misinformation of COVID-19 Vaccines and Vaccine Hesitancy. Sci. Rep. 2022, 12, 13681.
  • Kraśnicka, J.; Krajewska-Kułak, E.; Klimaszewska, K.; Cybulski, M.; Guzowski, A.; Kowalewska, B.; Jankowiak, B.; Rolka, H.; Doroszkiewicz, H.; Kułak, W. Mandatory and Recommended Vaccinations in Poland in the Views of Parents. Vaccin. Immunother. 2018, 14, 2884–2893.
  • Narodowy Instytut Zdrowia Publicznego – Państwowy Zakład Higieny Jaka jest liczba uchyleń dotyczących szczepień obowiązkowych? Available online: https://szczepienia.pzh.gov.pl/faq/jaka-jest-liczba-uchylen-szczepien-obowiazkowych/ (accessed on 7 May 2023).

Round 2

Reviewer 2 Report

Thank you for providing the chance to review the manuscript. My concerns have been addressed clearly.

Whileone more minor comment for the authors, in line 191, for the statistically significant P-value, it is used 0.05, 0.01 or 0.001, but not 0.005. 

Except for this, I don’t have further comments on it.

Author Response

Thank you so much for pointing out this mistake, and once again thank you for your kind review. 

Reviewer 4 Report

I appreciate the efforts in addressing my comments

I appreciate the efforts in addressing my comments

Author Response

Thank you so much once again for your kind review.